# Serological and Molecular Evidence of the Circulation of the Venezuelan Equine Encephalitis Virus Subtype IIIA in Humans, Wild Vertebrates and Mosquitos in the Brazilian Amazon

**DOI:** 10.3390/v14112391

**Published:** 2022-10-28

**Authors:** Franko A. Silva, Milene S. Ferreira, Pedro A. Araújo, Samir M. M. Casseb, Sandro P. Silva, Joaquim P. Nunes Neto, Jannifer O. Chiang, José W. Rosa Junior, Liliane L. Chagas, Maria N. O. Freitas, Éder B. Santos, Leonardo Hernández, Thito Paz, Pedro F. C. Vasconcelos, Lívia C. Martins

**Affiliations:** 1Department of Arbovirology and Haemorragic Fevers, Evandro Chagas Institute, Ananindeua 67030-000, PA, Brazil; 2Post-Graduation Program in Virology, Evandro Chagas Institute, Ananindeua 67030-000, PA, Brazil; 3Department of Pathology, Pará State University, Belém 66045-315, PA, Brazil

**Keywords:** arbovirus, metagenomic, Mucambo virus

## Abstract

Understanding the interaction between viruses and ecosystems in areas with or without anthropic interference can contribute to the organization of public health services, as well as prevention and disease control. An arbovirus survey was conducted at Caxiuanã National Forest, Pará, Brazil, where 632 local residents, 338 vertebrates and 15,774 pools of hematophagous arthropods were investigated. Neutralization antibodies of the Venezuelan Equine Encephalitis virus, subtype IIIA, Mucambo virus (MUCV) were detected in 57.3% and 61.5% of humans and wild vertebrates, respectively; in addition, genomic fragments of MUCV were detected in pool of *Uranotaenia (Ura.) geometrica*. The obtained data suggest an enzootic circulation of MUCV in the area. Understanding the circulation of endemic and neglected arboviruses, such as MUCV, represents an important health problem for the local residents and for the people living in the nearby urban centers.

## 1. Introduction

Arboviruses are mainly zoonotic viruses that are maintained in enzootic transmission cycles. Humans are considered accidental hosts, and the risk of infection is directly associated with exposure to environments where the viruses circulate [1,2,3].

The Brazilian Amazon forest has a large diversity of ecosystems. In addition to the climatic characteristics, the environment provides the necessary conditions for the occurrence and maintenance of transmission cycles of several arboviruses, some of which are agents of public health importance due to their capacity to cause diseases in humans [2,3,4].

Among these arboviruses, the *Mucambo virus* (MUCV) was isolated in 1954 in the Oriboca forest, Amazon region, Pará state, Brazil. Since then, MUCVs have been considered endemic to the Amazon and occasionally have been detected in other Brazilian regions and in French Guiana, Suriname, Trinidad and Tobago [5,6,7]. 

Taxonomically, MUCV is a member of the *Togaviridae* family and the *Alphavirus* genus. It is classified as subtype IIIA of the *Venezuelan Equine Encephalitis virus* (VEEV) and differs from subtypes I and II, which are epizootic and periodically cause outbreaks. MUCV is enzootic and actually has not been associated with outbreaks; rather, it has been responsible for sporadic occurrences of mild and limited febrile illnesses [5,6,7].

The association between high densities of hematophagous mosquitoes involved in the transmission cycles of several arboviruses [4,8] and the scarce population that is in close contact with the sylvatic environment highlights the importance of arbovirus studies in the Caxiuanã National Forest. Given the potential of the *Venezuelan Equine Encephalitis virus* complex to cause neurological conditions, it is extremely important to monitor the circulation of these viruses in human populations that live in forests and are exposed to the transmission cycles involving vectors and wild vertebrates. Thus, in the present study, we investigated the occurrence of the MUCV transmission cycle within the Caxiuanã National Forest environment by investigating all components of the cycle.

## 2. Materials and Methods

### 2.1. Place of Study

The Caxiuanã National Forrest is located in the State of Pará, Marajó region in northern Brazil (Figure 1). It is considered a highly preserved forest with a scarce population distributed in small communities.

### 2.2. Population 

For the human studies, blood was obtained between November 2014 and March 2016 from 632 individuals residing in 16 isolated communities who received information on the study and signed the informed consent. All studied people were more than two years of age, had lived in the area for at least one year prior to the study, and had no reports of chronic disease.

During the ecological study, a total of 15,774 hematophagous arthropods and 336 wild vertebrates were captured in 6 areas of the forest within the vicinities of the investigated communities between 2014 and 2016. The arthropods were captured using the CDC Light Trap, both in the soil and in the canopy. The wild vertebrates were captured using Shermann and Tomahawk traps.

### 2.3. Ethical Aspects

The study was approved by the Ethics committee on animal use of the Evandro Chagas Institute (CEUA/IEC) and the Chico Mendes Institute for Biodiversity Conservation (ICMBio) under reports 27/2016 and 42761-1/2014, respectively. All the individuals investigated signed the Free and Informed Consent Term form approved by the Ethics Committee in Research with Humans (CEP) of the IEC in 24th February 2016 under report number 1,942,982.

### 2.4. Serological Studies

Both human and sylvatic animal serum samples were screened using the hemagglutination-inhibition (HI) test [9,10]. Sera were tested against four antigens of Alphavirus genera *(Eastern Encephalitis Equine virus*, EEEV; *Western Encephalitis Equine virus*, WEEV, *Mayaro virus*, MAYV; *Mucambo virus*, MUCV), and HI antibodies were determined by detection of titles ranged from ≥1.20–≥1.1280.

A mouse neutralization test was performed on newborn (2–4 day old) Swiss mice (*Mus musculus*) [11] using the MUCV strain, BeAN 10,967. The LD_50/0.02 mL_ and logarithmic neutralization index (LNI) values were determined, and MUCV-neutralizing antibodies were determined with LNI ≥ 1.7 [12].

### 2.5. RT-PCR Screening

Mosquito pools were macerated in TissueLyser II (Quiagen^®^ Hilden, Germany). Genetic material was extracted using a Maxwell^®^ LEV 16 simplyRNA Tissue Kit (Promega, Madison, WI, USA) and quantitated using the Qubit™ 2.0 Fluorometer (Invitrogen™, Carlsbad, CA, USA ). For reverse transcription, the EasyScript^®^ First-Strand cDNA Synthesis SuperMix kit (TransGen Biotech, Beijing, China) and Anchored OligoDT reagent with poly A tail annealing and Platinum™ Taq DNA Polymerase (Invitrogen™, Carlsbad, CA, USA) were used. Specific primers were used for members of the Alphavirus genus for the NS1 region [13].

### 2.6. Nucleotide Sequencing, De Novo Assembling and Phylogenetic Tree

Positive samples were identified in RT-PCR and were sequenced by Next Generation Sequencing (NGS). The RNA was converted to cDNA using the cDNA Synthesis System Kit (Roche Diagnostics, Rich-Rotkreuz, Switzerland) and 400 μM of Roche “random” Primer. The reaction was purified with Agencourt AMPure XP Reagent (Beckman Coulter, Pasadena, CA, USA). The cDNA library was prepared and sequenced using the methodology described in the Nextera XT DNA Library Preparation Kit on a MiniSeq (Illumina, Inc., San Diego, CA, USA) platform that used a paired-end methodology with 300 cycles (2 × 150) in the Evandro Chagas Institute, Ministry of Health, Brazil.

The IDBA-UD v.1.1.3 software [14] was used to assemble all reads by a De Novo Assembler. All contigs were aligned and compared to the NCBI database of virus proteins, RefSeq, available through the Diamond v. 2.0.15 software [15]. The inspection of the viral contigs and annotations of putative open reading frames (ORF) of genes was performed using the Geneious v.9.1.6 software (https://www.geneious.com Biomatters, New Zealand, accessed on 10 July 2019).

Initially, multiple sequence alignments were performed using the MAFFT v.7 software [16]. The jModelTest v.2 software was used to identify the best model-fit analysis of nucleotide sequences [17]. The maximum likelihood method was used to construct the phylogenetic tree [18] with the RaxML v.8.2.4 software [19], using the non-structural and structural polyprotein of different Alphavirus and MUCV strains available in the NCBI. The bootstrap was performed in 1000 replicates to determine the reliability of the tree [20]. The phylogenetic tree was visualized using FigTree v.1.4.4 software (http://tree.bio.ed.ac.uk/software/figtree/, accessed on 29th August 2019). 

### 2.7. Statistical Analysis

The spatial distribution map was generated with the ArcGis v.10.0 software (Redlands, CA: Environmental System Research Institute, Inc., 2010), and the geo-statistics analysis was performed using GeoDa v.1.20 software [21]. Spatial autocorrelation was the statistical test applied, and the Moran dispersion diagram [22,23] indicated the degree of spatial dependence. Finally, a cluster analysis was performed with the aim of grouping similar locations together.

## 3. Results

All of the 632 individuals investigated were asymptomatic during the study. No significant differences were observed between the sexes. Regarding age, the population was young, and the age group with the highest frequency of investigations was younger than 20 years old, 359/632 (56.8%). A significant number of individuals, 479/632 (75.6%), reported frequent monthly travel between the communities and surrounding cities.

In the HI assay on human serum samples, 147/632 (23.3%) of the samples presented antibodies to members of the *Alphavirus* genus, and 89/147 (60.5%) had antibodies to MUCV (Table 1).

In the ecological study, a total of 63 vertebrate serum samples were obtained, and all originated from animals of the Mammalia class belonging to the order Rodentia. In the HI assay, 13/63 (20.6%) serum samples presented monotypic reactions to MUCV. 

A mouse neutralization test was performed in all human sera samples that presented MUCV antibodies, and 51/89 (57.3%) of the tested samples presented neutralizing antibodies against MUCV. Of these, 27/51 (53%) were from males and 24/51 (47%) from females. In males, a slight predominance was observed for young people under the age of 18 (55.5%); in the female group, no difference was observed regarding age.

In the wild vertebrates, neutralization antibodies were confirmed in 8/13 (61.5%). Of those, 6/8 (75%) were from rodents belonging to the genera *Oecomys* sp., 4/6 (66.6%), and *Proechimys* sp., 2/6 (33.4%). 

To evaluate the spatial relationship between the communities and the presence of residents with neutralizing antibodies to MUCV, we used the Moran scatter diagram (Figure 2).

Among the 16 communities investigated, 12 were found to have inhabitants with neutralizing antibodies to MUCV (Figure 2A). In this sense, the Moran scatter diagram was used to indicate the degree of spatial dependence and statistical significance by verifying the existence of spatial autocorrelation and the presence of spatial patterns (Figure 2B).

The value presented by the Moran scatter plot (v = 0.4778463) indicated a strong positive spatial autocorrelation, demonstrating a spatial association between the localities. The analysis of clusters revealed the presence of four types of clusters: High-High, Low-Low, Low-High and High-Low (Figure 2C).

During the entomological study, 15,774 mosquito specimens were collected and classified into 56 species and 867 pools for virus investigation partial MUCV genome was detected in only one batch (0.11%) of the *Uranotaenia (Ura.) geometrica* species by RT-PCR screening, and complete genome of the all coding genes region was obtained. Analysis using preexisting sequences in GenBank revealed that the MUCV sequence had a high identity with other MUCV and *Tonate virus* (TONV) genomic sequences from NCBI, under the RefSeq ID NC038672 and NC038675, respectively (Table 2).

The phylogenetic tree of MUCV (BeAr816662), with several other samples from the Alphavirus family, showed a monophyletic clade inside of the Venezuelan Equine Encephalitis virus complex (Figure 3a). A phylogenetic inference using only MUCV strains revealed three well-defined clades (Figure 3b). The clade 1 isolates comprised the *Culex* (*Mel*.) *portesi* species from Trinidad and Tobago, obtained in the years 2007 and 1965 [24]. In the clade isolates, samples also showed origins in Trinidad and Tobago but from rodents and sentinel mice, and only two samples were from mosquitoes (TRLV115847 and TRLV62655) obtained in the decade between the 1960s and 1970s. The clade 3 isolates comprised samples isolated in Brazil, identified between the years 1954 and 1969, and within the time of the collected samples from this study, but in an external branch to the group. The non-synonymous mutations in the polyprotein structure showed a correlation with three clades (Figure 3c).

## 4. Discussion

Currently, in the Americas, ten arboviruses have been associated with encephalitogenic disease in humans and domestic animals, of which three belong to the family Togaviridae and the Alphavirus genus (*Eastern Equine Encephalitis virus*—EEEV, *Western Equine Encephalitis virus*—WEEV and *Venezuelan Equine Encephalitis virus—VEEV*). However, a large number of human infections are inapparent or subclinical and are commonly manifested by an undifferentiated and benign febrile illness that affects all age groups [3,5,7,25]

The prevalence of MUCV antibodies can reach up to 35% for some populations in the Brazilian Amazon. Despite this, MUCV is considered a neglected arbovirus. This may have occurred due to it being a mild, self-limited disease, its relative geographic isolation of some Amazon populations, or a lack of active and systemic surveillance and proper laboratory diagnoses [2,25]. All of these factors may play a role in the few numbers of annual reports of the human disease [2,25] and potentially underestimate MUCV epidemiological importance, not only in isolated populations but also in the entire population of the Brazilian Amazon region.

MUCV is enzootic in the Pan-Amazonian region, and it has been isolated in Brazil, French Guiana, Suriname, Trinidad and Tobago [3,5,6,7,24,25,26]. Mammals of the order Rodentia, especially the *Oryzomys capito* and *Proechimys guyannensis* species, are considered amplifier hosts for MUCV [5,6,7]. In this study, the detection of neutralizing antibodies indicated that these rodents were exposed to MUCV within the investigated area. Although, the sole detection of neutralization antibodies is insufficient to determine that these rodents act as local amplifier hosts.

Several studies incriminated *Culex (Mel.) portesi* as the main vector of MUCV, but other species, such as *Ae. (Och.) hortatory*, presented as a potential vector [3,5,7,26]. The detection of the MUCV genome in a pool of the geometric *Uranotaenia (Ura.) geometrica* species is interesting and unexpected in the observed epidemiological pattern since there are no previous reports of MUCV being isolated from that species, and there is no epidemiological information on the importance of the species in the arbovirus transmission. Further studies are necessary to evaluate the potential involvement of the species as a competent vector in the maintenance and dispersion of MUCV in the FLONA of Caxiuanã and surrounding municipalities.

For some arboviruses, a higher prevalence of infection in males is commonly observed due to higher exposure [1,2,3,25]. In our study, we observed the same pattern, which may be related to the socioeconomic factors that result in increased exposure; fishing and hunting are important subsistence activities carried out by these populations. In addition, fishing is an important recreational activity exercised primarily by young men [27,28]. 

The sequence recovered in this study had a high identity with other MUCV sequences and other subtypes of VEEV, especially the TONV classified as the VEEV IIIB subtype. In the phylogenetic analyses, the sequence had a closer proximity to the common ancestor of MUCV than other sequences of MUCV that originated in Brazil. However, the sample stayed isolated in a branch of the tree that tried forming an external independent branch. Therefore, it is necessary to sequence more MUCV in the same region of this sample to determine the resolution of the Brazilian clade.

In the spatial distribution, we observed the presence of clusters with strong spatial associations between the communities and the presence of residents with MUCV-neutralizing antibodies. The Moran scatter diagram helped verify that these clusters were not restricted to a delimited area within the Caxiuanã National Forest, suggesting that a silent MUCV transmission cycle is active inside the entire area. Interestingly, four different clusters were recognized, and we hypothesize that the transmission cycle is not similar in all communities. Indeed, they may not occur in close proximity to all of the investigated communities, and residents with neutralization antibodies to MUCV that lived in communities without a spatial association have probably been exposed due to the frequent travel within the area and the surrounding cities. 

The detection of Alphavirus antibodies in the investigated population without MUCV-neutralizing antibodies suggests the circulation of different Alphaviruses in the region. Further studies are necessary to address this question since other Alphaviruses, such as EEEV, WEE and MAYV, are important public health concerns [4,24,26].

The frequent travel between the communities and nearby cities and the high density of mosquitoes may represent an important public health problem. A more extensity epidemiological surveillance is necessary to assess the arbovirus diversity in Caxiuanã, as well as the impact of those viruses on the local population and for more than 20 million Brazilians living in the Amazon region.

## Figures and Tables

**Figure 1 viruses-14-02391-f001:**
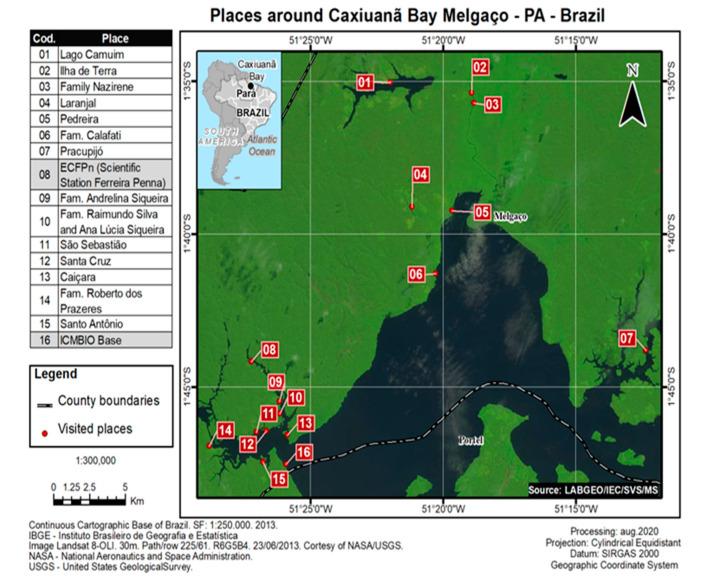
Description of the studied areas in the Caxiuanã National Forest.

**Figure 2 viruses-14-02391-f002:**
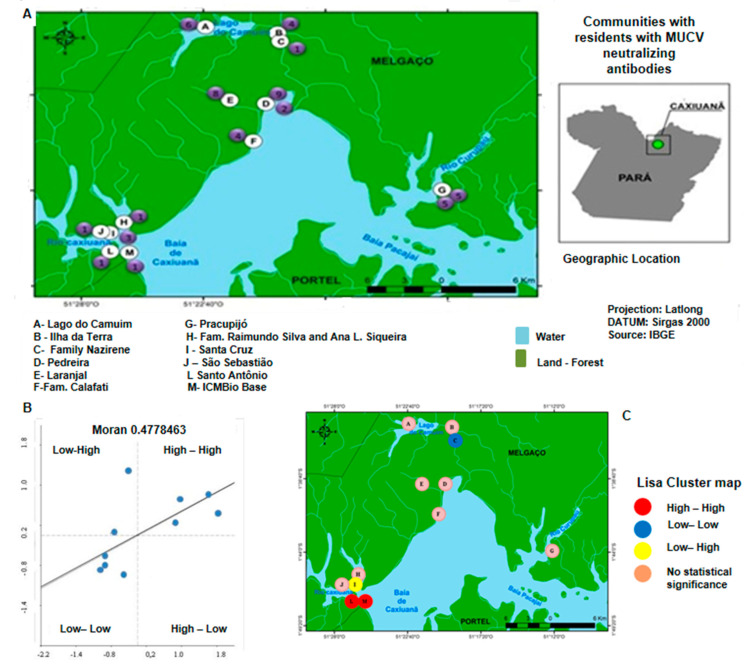
(**A**) Communities where residents presented neutralizing antibodies to MUCV. (**B**) The Moran scatter diagram. (**C**) Clusters. The number inside the purples spots (A) represents the number of inhabitants per community with neutralizing MUCV antibodies. The Moran scatter diagram (B) indicates a strong spatial autocorrelation. The analysis of clusters (**C**) revealed the presence of four types of clusters: High-High; Low-Low; Low-High; High-Low.

**Figure 3 viruses-14-02391-f003:**
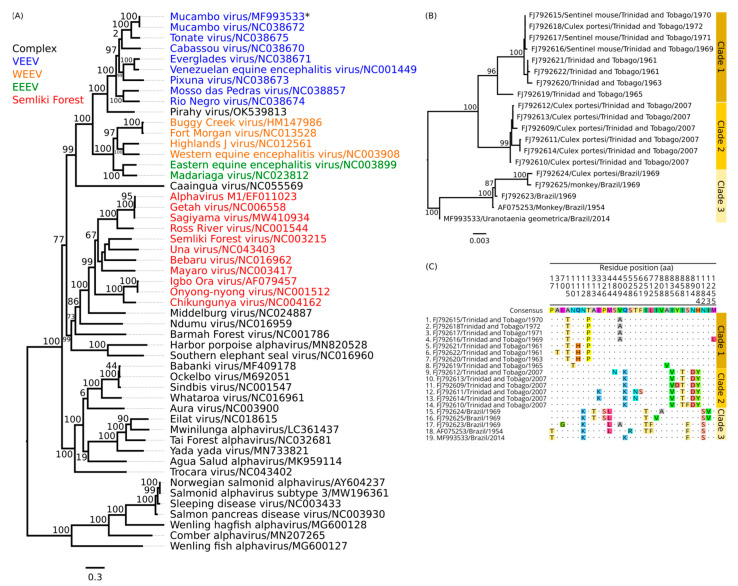
Analyses using different sequences of Alphavirus. (**A**) Phylogenetic tree of different RefSeq Alphaviruses using the Maximum Likelihood (ML) based on the complete nucleotide sequences of a non-structural polyprotein. The GTR + F + I + G4 distribution was the best-fit model defined for this dataset. The tips of the tree correspond to the NCBI ID/Strain/Host/County/Year of each sample. (**B**) Phylogenetic tree using partial nucleotide sequences of structural polyprotein of a region with 3777 nt. Three monophyletic groups are identified. For both trees, the numbers at each main node of the tree correspond to bootstrap values in percent (1000 replicates). The scale bar corresponds to genetic divergence among nucleotide sequences. (**C**) Amino acid alignment of the non-structural polyprotein, including only non-synonymous mutations, and the position in the top image correspond to the position in the MF993533 sequence available in the NCBI.

**Table 1 viruses-14-02391-t001:** Serological results, detection of *Alphavirus* and MUCV antibodies (HI), and MUCV neutralizing antibodies in the investigated population.

	Human	Wild Vertebrates
Hemagglutination inhibitor antibodies to the *Alphavirus* genus	147/632 (23.3)	13/63 (20.6)
Hemagglutination inhibitor antibodies to MUCV	89/147 (60.5)	13/63 (20.6)
Neutralizing antibodies to MUCV	51/89 (57.3)	8/13 (61.5)

**Table 2 viruses-14-02391-t002:** Identity matrix of MUCV with other alphaviruses using the concatenate non-structural and structural polyprotein sequence.

		1	2	3	4	5	6	7	8	9	10
1	*Mucambo virus*/NC038672		98.9	82.7	74.1	72.9	73.3	72.0	70.7	70.6	57.4
2	*Mucambo virus*/BeAr816662	99.6		83.1	74.1	72.9	73.1	72.0	70.8	70.6	57.4
3	*Tonate virus*/NC038675	92.9	93.0		73.7	72.9	73.4	71.5	71.0	70.3	57.7
4	*Venezuelan Equine Encephalitis virus*/NC001449	83.6	83.6	83.5		88.2	72.5	73.0	71.5	70.5	58.6
5	*Everglades virus*/NC038671	84.3	84.6	83.4	82.3		72.1	72.3	70.3	70.5	57.6
6	*Cabassou virus*/NC038670	83.5	83.6	82.7	81.3	94.8		70.5	70.4	70.1	58.2
7	*Pixuna virus*/NC038673	80.0	80.0	79.1	80.0	80.1	79.4		70.5	69.6	58.0
8	*Rio Negro virus*/NC038674	77.7	77.5	78.3	77.5	78.2	77.6	77.4		74.1	57.9
9	*Mosso das Pedras virus*/NC038857	78.2	78.3	77.8	77.3	79.4	78.2	77.8	83.6		57.4
10	*Madariaga virus*/NC023812	56.9	56.8	56.3	56.1	56.4	56.3	55.6	55.8	56.1	

The gray area highlights the highest identity, and values in the top right half of the table show nucleotide identity, and the bottom left half of the table show amino acid identity.

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
