# Peer review of "Serological and Molecular Evidence of the Circulation of the Venezuelan Equine Encephalitis Virus Subtype IIIA in Humans, Wild Vertebrates and Mosquitos in the Brazilian Amazon"

_viruses, 2022, doi:10.3390/v14112391_

Round 1

Reviewer 1 Report

The work is methodologically sound, I do not see major flaws in the interpretation of the data.  The final message in the abstract needs to be toned down, and in particular “The emergence of human infections of MUCV and the risk of virus outbreak is of concern and represents an important health problem for the local residents, and for the people living in nearby urban centers. ” since no outbreak of MUCV has been described so far.

Figure 2 doesn’t make sense. Can authors please represent it as a table instead? Or explain what they were traying to achieve? I don’t think it’s ok to keep this figure. Also, figure legend it’s too basic, if some information would have been added here it might have been easier to understand the meaning of this figure.

Minor comments:

Quality of the figures needs to be improved, right now they are blurry.

Please fix legend of figure 3. Do not need to add “legend” in the legend.

Revision of English needs to be done, for example:

“For the human studies, 632 individuals residing in 16 isolated communities were bled 57 after receiving information on the study” to be bled sounds like a wound and great amount of bleeding. I would rephrase the sentence saying that blood was drawn from XXX people.

Please add to the sentence when the blood was obtained.

Reviewer 2 Report

In the manuscript entitled "Serological and molecular evidence of the circulation of the Venezuelan equine encephalitis virus subtipo IIIA in humans, wild vertebrates and mosquito in the Brazilian Amazon” the author described an arboviruses survey through investigated the collected specimens, including 632 local residents, as well as 338 vertebrates and 15,774 pools of hematophagous arthropods.

Major Compulsory Revisions 

1. The Amazon rainforest is considered the largest reservoir of arboviruses in the world. However, there is no more information about arboviruses and MUCV in the introduction.

2. It is lacking about details of the specimen in this paper. Suggest that it could be given in supplemental tables. For example: age, gender, residence and other information of residents; information about vertebrates and arthropods specimens, etc.;

3. How many genotype in MUCV? From the phylogenetic analysis in fig, how could you say the strain belonging to subtype IIIA.

4. The description of the result is confused and the hierarchy is not clear, and suggest the author to revise it.

5. This article lacks references for the past 5 years.

Minor Essential Revisions

1. Page 3 line 76: Mouse neutralization test, add MUCV strain information.

2. Page 4 line 120, 130: “IH assay” changed to “HI assay”.

3. Table 1. why selected these alphaviruses strains to do the identity. Are all these alphaviruses endemic in South America? Why not select WEE and/or Chikungunya and/or Mayaro virus?

4. There are two figure 3. The figure marking is error. 

Reviewer 3 Report

Mucambo virus is an alphavirus that is known to be circulating in South America without massive outbreaks yet. Not much information on the natural transmission cycle, potential spill over mechanisms and transmission vectors have been understood. The manuscript describes a prevalence study of MUCV using serology and molecular biology methods in the Amazon area. 

The study showed high prevalence of antibody positivity in the humans and rodents in the area, showing an active circulation cycle of MUCV. In addition, they were able to reconstitute a full genome virus sequence from a Uranotaenia sp., implying a novel transmission host for MUCV.

However several potential issues that need to be addressed are noticed 

1. Title : Subtipo IIIA should read ‘subtype IIIA’

2. 2.4 : The positive criteria is not presented. Detailed information on how serotypes were determined is important to evaluate “MUCV” prevalence. Can the test distinguish MUCV from other VEEV strains? If not, how can the study address the prevalence MUCV? 

3. Serology : While LNI and HI titers of the collected sera were determined, the results are not presented and a simple positive rate is described. Serology data would be informative to understand the active circulation of the virus in the area.

4. Results :

            1) Any histories on fibril illness among the subjects are available for those showed positive in serological tests?

            2) Animals are described “Mammalia class” and no species or genus information are provided. This information is important to understand the natural host of MUCV and should be included in the manuscript. For instance, sero prevalence in different species would show the natura host MUCV. 

            3) Figure 2. In addition to the % rate, no. of cases should be included to provide a better understanding.

            4) Line 158 : the sentence is incomplete.

Author Response

"Please se the attachment"
